# “Molecular Biology”—Pleonasm or Denotation for a Discipline of Its Own? Reflections on the Origins of Molecular Biology and Its Situation Today

**DOI:** 10.3390/biom13101511

**Published:** 2023-10-12

**Authors:** Gregor P. Greslehner

**Affiliations:** Department of Philosophy, University of Vienna, Universitätsstraße 7, 1010 Vienna, Austria; gregor.greslehner@univie.ac.at

**Keywords:** molecular biology, history, discipline, central dogma, philosophy of science

## Abstract

The disciplinary identity of molecular biology has frequently been called into question. Although the debates might sometimes have been more about creating or debunking myths, defending intellectual territory and the distribution of resources, there are interesting underlying questions about this area of biology and how it is conceptually organized. By looking at the history of molecular biology, its origins and development, I examine the possible criteria for its status as a scientific discipline. Doing so allows us to answer the title question in such a way that offers a reasonable middle ground, where molecular biology can be properly viewed as a viable interdisciplinary program that can very well be called a discipline in its own right, even if no strict boundaries can be established. In addition to this historical analysis, a couple of systematic issues from a philosophy of science perspective allow for some assessment of the current situation and the future of molecular biology.

## 1. Introduction

The main question of this paper is whether and why molecular biology can be considered a discipline of its own. (This paper is in parts a translation and adaptation of my 2012 bachelor’s thesis at the University of Salzburg under Michael Breitenbach’s supervision. When discussing my thesis with Michael, he asked me whether I had already published anything, which I had not at the time. Being occupied with other academic pursuits (including a master’s thesis in Michael’s team on the yeast NADPH oxidase gene *YNO1*), I never actually got to publish on the historical and philosophical trajectory of molecular biology. This special issue “in honor of Prof. Michael Breitenbach” is now a welcome opportunity to finally do so. Thank you, Michael, for your broad and thorough scientific and philosophical interests!) The answer that I develop and defend here is as follows. If by “discipline” one means a stringent and uniform theory or a closed and uniform framework of institutions, methods, theories, etc., then molecular biology would *not* be a discipline of its own. However, the same negative assertion would then also apply to numerous other domains of research that are usually considered disciplines without any controversy. Hardly any of the many fields, such as medicine, meteorology, geology, philosophy, etc., that we ordinarily do refer to as “disciplines”, would fall under such a narrow conception of what constitutes a discipline. We may thus rightfully ask what the criteria are to justifiably calling something a discipline. Ideally, such a notion would not be so narrow as to exclude its intended extension. On the other hand, when taking into account that in practically all areas of biological research molecular aspects play an essential role, one might ask whether the boundaries are totally blurred and whether any biology nowadays would be *molecular*. That is to say, “molecular biology” might be considered a pleonasm like “round circle”, “unmarried bachelor” or “dead corpse”. Putting “molecular” in front of “biology” would thus be verbally redundant.

Between those two extreme positions, pleonasm or clear-cut discipline of its own, one needs to find a balanced and substantially adequate position. This is what I attempt in this paper. By doing so, we will also have to ask the question whether thinking in disciplines is at all appropriate for today’s scientific network-landscape. Not only is this important for providing some orientation within the ever-growing landscape of the life sciences but it also provides lessons from the conceptual history and research tradition that molecular biologists are still part of today. In order to address this main question about the disciplinary identity of molecular biology, we will inevitably also touch upon some general questions and problems from the history and philosophy of science. We will do so by investigating the emergence of the field called “molecular biology” (whether or not we ultimately decide that it can be justifiably considered to be a discipline). Addressing this question is not merely a fight about semantics and which terms can be applied—these questions are crucial for navigating the present, past and future of molecular biology, upon which a number of scientific, historical and philosophical questions depend.

This paper will proceed as follows. First, I will argue in Section 2 why the history and philosophy of science are not only relevant to specialists from these fields in the humanities but also to those from the subject at hand, i.e., molecular biologists. Section 3 does a lot of heavy lifting by tracing the historical development of molecular biology in order to eventually answer the title question about the disciplinary status of molecular biology. The answer will be to regard molecular biology neither as a pleonasm nor as a discipline in a narrow sense. Instead, I suggest a wider notion of discipline, in which the central aspect of molecular biology’s interdisciplinarity can be embraced as a defining feature. This will be achieved by comparing a number of different positions on the disciplinary status of molecular biology and its development, by addressing the peculiar formulation of the “central dogma of molecular biology”, and by developing a nuanced position based on these historical details. Finally, Section 4 closes with some reflections on the current situation and future developments of molecular biology.

## 2. Why History of Science Matters

Whereas it would be difficult to find anyone openly denying the worth of history of science in general, there are mixed opinions about its value for the everyday practice of science. Why should someone working at the lab bench have to deal with old chestnuts or dig up old, dusty journals and lab notebooks? One might say: “Shouldn’t we push the frontiers of science instead of stirring up old stories and outdated ideas that didn’t hold up to today’s scientific standards? There are better things to do, researchers are busy surviving in their precarious publish-or-perish environment anyway. Let the historians and philosophers have their fun—at least they won’t be in the way and prevent us from doing *real* science”.

Such an attitude would certainly be detrimental to science and its progress. Just imagine that no one ever cared about the results from a certain Augustinian monk who bred peas in his garden. However, apart from such spectacular examples such as the rediscovery of Mendel’s observations by de Vries, Correns and Tschermak in 1900 (but see [1])—which perhaps should have solidified a different attitude towards history in biology—the history of science and the practice of science rarely ever go hand in hand. As Erwin Chargaff remarks, “most papers cite almost nothing older than five years” [2] (p. 193, my translation). However, there is hope that historical roots do get some attention from specialists in the field. Unfortunately, however, most of the time it remains at the level of a few interspersed names, anecdotes, and curiosities: “This is where many scientists begin with history—with an interest in anecdotes and a sketch of the background to current work. Unfortunately, most scientists end there” [3] (pp. 344–345).

This is unfortunate, as historical analysis and rational, logical reconstruction of the process of knowledge generation is a core issue in the philosophy of biology. The Austrian biologist F. M. Wuketits even opined: “Contrary to common opinion, I am convinced that no consideration of modern biology can do without resorting to history” [4] (p. VI, my translation). W. Nachtigall critically remarks that “biology is not history of biology, whereas philosophy is at least four out of five parts history of philosophy” [5] (p. 130). Despite this piece of polemic criticism, Nachtigall’s writing is a prime example of how philosophical considerations can be very fruitful for biological reasoning. Perhaps he had in mind certain forms of “philosophy” that indeed have often not been very helpful or interesting for scientists.

S. G. Brush raises the question whether the distortions from history of science might actually be harmful, especially for students entering a discipline: “Should the history of science be rated X?” [6]: “I will examine arguments that young and impressionable students at the start of a scientific career should be shielded from the writings of contemporary science historians for reasons […] that these writings do violence to the professional ideal and public image of scientists as rational, open-minded investigators, proceeding methodically, grounded incontrovertibly in the outcome of controlled experiments, and seeking objectively for the truth, let the chips fall where they may” [6] (p. 1164).

The criticism might be somewhat anachronistic and aimed at the wrong target. However, perhaps such considerations are at least part of the reason why history of science usually has no place in the curricula of individual disciplines. This is unfortunate, as it would allow for a better understanding of how science works. “Scientific development is like Darwinian evolution, a process driven from behind rather than pulled toward some fixed goal to which it grows ever closer” (T. S. Kuhn 1992 *The Trouble with the Historical Philosophy of Science*, cited after [7] (p. 139)).

One way in which history of science can benefit scientists is also by letting go of some illusions: “the delight of demystifying science of its more pretentious claims to rationality and orderly procedure, the delight of revealing for once, or seeming to reveal, what scientists “really do”—which turns out to include coffee breaks and gossip” [8] (p. 26).

Maienschein and colleagues distinguish at least five general categories in which the history of science matters for scientists:“Self-improvement, illuminating science and making it betterEfficiency, avoiding and learning from past mistakesPerspective, providing judgment and clarity and therefore making science betterImagination, offering a wider repertoire of ideas to choose fromEducation, improving public understanding of science and scientific literacy” [3] (p. 342).

There is some awareness of the importance of history in contemporary research and teaching practice, as a welcome example from a recent textbook shows: “we have continued to present the field in a historical context, with the intent of sensitizing and inspiring students (and others) to the realities of how research progress unfolds and how ideas develop and attain maturity—or not. We have refrained wherever possible from unadulterated dogma and from presenting the field […] as anywhere near total clarification. While we are aware of presenting viewpoints that are sometimes controversial and even conflicting, we trust that readers, especially students, are not unduly confused or frustrated by our reluctance to always provide the final word, as it were. Rather, it is our hope that such controversies and complexities will inspire further studies” [9] (pp. xxv–xxvi).

Historical considerations seem to be the common element in most philosophical discussions about molecular biology (cf. the extensive reference list on the history of molecular biology in [10]). Taken together, there has been a rather large interest in the history of molecular biology, although it has subsided a bit recently. (It appears paradoxical, as S. de Chadarevian observes that “steady advances in the field of molecular biology” coincide with “the fading interest in its history” [11] (p. 464).) However, there are still a number of chapters left to process. The aim of this paper, however, is neither a complete overview nor a particularly original compilation of this history. One may even question whether that is possible at all: (“The history of the past can be described—most often wrongly—but I do not think it can be explained; or perhaps better, there are too many different explanations that are equally valid and equally useless” [12] (p. 239, my translation).) “Unfortunately I have concluded that there cannot be a history of natural science that would be more profound than, let’s say, a history of fashion” [12] (p. 131, my translation). In order to still make good use of history, I think it is best to openly acknowledge one’s necessarily limited perspective. “In truth, history always appears to me to be rather subjective, arbitrary and selective. And whoever engages in historical reflections would be maybe well advised to drop any claims to objectivity and instead at least give one’s criteria for their arbitrary and subjective selection of topics and their treatment and to disclose what one’s aims are” [13] (p. XI, my translation).

To be transparent about my aims here: I want to argue that studying the history—not only of one’s own discipline—enriches the everyday business of systematic researchers. In doing so, one should not only consider the already established and easily accessible resources but also those that might have received less attention. Who knows what kind of treasures are still buried in the archives? The value that the history of molecular biology has to offer—in addition to the general reasons outlined above—also lies in the fact that biology is a dynamic and rapidly evolving domain of research. In order not to float aimlessly alongside the latest trends without any orientation, studying some history provides an important perspective. Finally, there is the title question of this paper: where to define the place of molecular biology as a scientific discipline?

## 3. “Molecular Biology”—Pleonasm or Denotation for a Discipline of Its Own?

“The biggest difficulty in discussing the history of molecular biology is in knowing just what the term means, and what lines of demarcation can be drawn between it and other fields of biology. Of course biology is multidisciplinary, and its area of interest lies at the interface between several established fields […] and the question arises what distinctive element there is in the approach of molecular biologists to justify a special name” [14] (p. 5).

Before tackling the multi-faceted question about molecular biology’s status as a scientific discipline, let us start with the most basic question: what is molecular biology?

### 3.1. What Is Molecular Biology?

“What is *x*?” questions are often the most difficult to answer and often how philosophical debates get started. When asked by family or friends what one’s area of scientific work centers around, one often has to first clarify that “microbiology” and “molecular biology” are different terms and different fields. With that out of the way, you might then have to explain to your increasingly confused conversation partner that it is a kind of mixture of biology, physics, chemistry, genetics, etc., that studies the processes of living organisms on the molecular level. You may go on to highlight, for example, the particular protein or pathway your lab is working on—and how it might in the future contribute to understanding a certain disease. Frequently, that will be acknowledged with some interest and skepticism. Sooner or later, however, the inevitable question on an afternoon walk will come up: “which kind of plant or animal is that? Aren’t you a biologist? You should know!” Unless by lucky chance you can answer this question, your opposite will feel confirmed in their skepticism. “This one wants to be a biologist? They don’t even know the animals/plants (cf. [15] (p. 14, my translation))! No surprise, if they study everything mixed together without *really* knowing any of these scientific disciplines”.

A very different experience will unfold when you have to explain to the same person something about CRISPR genetics that was mentioned on the news. However, the impression will remain: molecular biology appears to be neither fish nor fowl. Is this impression entirely unwarranted? Or does it hit a meaningful weak spot? Or, perhaps, might it actually be one of molecular biology’s assets? A look at the history and development of molecular biology and the origin of its name will allow us to answer these questions and shed light on some of its central theoretical features.

The main question of this paper is whether molecular biology is a scientific discipline of its own. Since the answer seemingly could not be anything other than “obviously, yes—duh”, the question might appear ridiculous and absurd. Surely there are departments, study programs, scientific journals and societies bearing that name. Whether or not that is indeed a good criterion is a separate question. However, a closer inspection of such study programs and the term “life sciences” in general reveals some interesting observations. The usage of the plural form “scienc*es*” seems to “suggest that a number of (still?) heterogeneous areas are assembled under this umbrella term” [16] (p. 115, my translation). Additionally, looking into the curricula of study programs does not contribute to the impression of molecular biology as a coherent and stringent discipline, as you will find courses on chemistry, physics, genetics, mathematics, cell biology, biochemistry, biophysics, even biophysical chemistry, bioinformatics, molecular medicine, etc.

Thus, one could easily dispute that molecular biology is a discipline, as a subject with its own methods and issues. Others, on the other hand, might equally feel inclined to say that “every piece of genetic or other biological research contains some part of molecular biology, because ultimately everything can be traced back to the changes of molecules” [5] (pp. 43–44, my translation). However, Nachtigall eventually concludes that “not for any epistemological reasons, but due to purely practical aspects, biology is a closed discipline” [5] (p. 49, my translation), and further: “Biological questions are often characterized by the fact that in order to answer them a diversity of different methods have to be combined in multiple ways. This fact makes modern biological research so difficult” [5] (p. 50, my translation).

Thus, the question “whether the life sciences constitute a *logical whole*” [17] (p. 181, my translation, original emphasis) deserves more attention. In any event, it seems to be a fact when Sarkar finds that “[t]oday, for good or for bad, depending on one’s point of view, most of biology is molecular biology” [18] (p. 1). However, is that really the case for all of biology? How does molecular biology relate to the rest of biology? How does it relate to other scientific disciplines such as physics and chemistry? How did this relationship evolve? What are the possibilities and dangers of such relationships? What can philosophy of science learn from these developments—and contribute to them? These are a number of questions to which a historical and philosophical analysis of molecular biology can contribute. In return, molecular biology poses a prime study object for the history and philosophy of science to study topics of, e.g., theory change and dynamics. *Interdisciplinarity* is another important aspect. From an inter- and transdisciplinary point of view, it might not be a flaw at all that the status of molecular biology as a scientific discipline can be called into question. To the contrary, it might be a sign of healthy interdisciplinary practice.

If we were to take the name of a field of research as a main indicator for its status as a discipline and considering all the practical and organizational reasons for doing so, it clearly cannot be the only factor. Along the same lines, it seems questionable whether the brackets holding molecular biology together have been consistent enough to allow its disciplinary identity through time: “molecular biology had a strong disciplinary identity perhaps only in the fifteen years or so following the discovery of the structure of DNA. […] In saying that molecular biology had a strong disciplinary identity we mean that the name became, for a time at least, associated with recognisable clusters of concepts, projects, tools, institutions, and individuals” [19] (p. 62).

Perhaps then it will be better to call molecular biology an “ultra-discipline” [20] (p. 4) or “super-discipline” [21] (p. 79)—something overarching and uniting parts of other disciplines—or to simply accept a less narrow sense of “discipline”, in which we do not require strict boundaries. In any event, it is clear that we have to reconsider the concepts of and requirements for the identity and development of scientific disciplines (for a case study about molecular evolution, see [22]). This is especially true for molecular biology, for which R. M. Burian has the following fitting claim: “there is not, I claim, a central theory that binds together the many distinct (sub)disciplines of molecular biology. What serves to unify the distinct sorts of work in molecular biology together is *not* a central theory, but the use of an immense battery of techniques and a general approach to explaining—and altering—organismic function by reference to, and use of, an *omnium gatherum* of detailed molecular mechanisms.

In brief, molecular biology is less like Newtonian mechanics than it is like auto mechanics.

Despite the lack of an overarching theory, a Newtonian or quantum mechanics of its very own, molecular biology has become a unifying discipline in virtue of the powers of its techniques, its ability to extrapolate from the molecular to higher levels, and its synthesis of problems of form and function at the molecular level. This synthesis of form and function is a central, ill-understood, and historically important feature of molecular biology. Among other things it explains the degree of truth—and of error—in Erwin Chargaff’s and Seymour Cohen’s mordant witticism that molecular biology is the practise of biochemistry without a license” [21] (pp. 67–68, original emphasis).

Finally, it is insightful to emphasize that the question of molecular biology as a discipline might sometimes have been confused with the question of molecular biology *as a theory*. (The “central dogma” as a potential candidate for such a (theoretical) role is discussed later. Its assumptions “may well be taken to form the theoretical core of molecular biology (to the extent–and this is a matter of controversy–that molecular biology has any theory)” [23] (p. 187).) Unlike other areas of science, however, the notion of “theory”—understood as a deductively closed set of sentences—might not be suited for most areas of biology (for a seminal study on theory structure and change in molecular biology, see [24]). If not through a central theory, how could something be considered a scientific discipline? We will address this question next.

### 3.2. What Is a Scientific Discipline?

Following B. Gräfrath’s encyclopedia entry, *scientific discipline* is the “denotation for an area of science that can be demarcated by its subject matter, method or epistemic interest” [25] (p. 237, my translation). Now, is molecular biology such an area of science that can be demarcated by its subject matter, method, or epistemological interest?

The subject matter encompasses all processes and entities of the living world, with special regard to their molecular features. What would set molecular biology apart from the rest of biology other than this *special regard*? Moreover, would it suffice as a demarcation criterion (especially against biochemistry)? Other areas of biological research with different aims often pay an equal special regard to molecular features. On the other hand, molecular biology is not only interested in and limited to molecular features; the macroscopic, cellular, and physiological levels also play an important role in molecular biology all the time. Without going further down the rabbit hole of the notions of levels and their relations [26], we can see that subject matter alone does not allow for a satisfactory demarcation of molecular biology.

Perhaps we fare better with the method, consisting of the “immense battery of techniques” Burian observed above to provide a constitutive element for molecular biology through its methods. Among the methods that might put some shape to the body of molecular biology are indeed a whole arsenal of techniques. Without claiming to give a remotely complete list, these include: PCR, cloning, sequencing, gel electrophoresis, X-ray crystallography, fluorescence in situ hybridization, Southern blots (and all other cardinal direction variations), and many more. Although it does warrant specialized training, education, and awarding academic degrees, this bandwidth will most likely not work to firmly demarcate molecular biology as its own discipline either. (It is an interesting observation (thanks to an anonymous reviewer) that the name “molecular biology” has increasingly disappeared from study programs and textbooks, where some of the most influential ones are titled “Molecular Biology of the Cell” [27] and “Molecular Cell Biology” [28]).

Finally, to our third and last ingredient. The epistemic interest might in fact offer the best justification for the term “molecular” in this denotation for an area of biology. What falls into the epistemic interest, however, is less a question of curricula and academic teaching but rather depends on the research interests of individual researchers. In today’s molecular biology, the interests of people working in the field are perhaps even more diverse than what we normally consider to be within the scope of a given discipline. For that reason, I will leave the discussion about the current status and future trajectory of molecular biology for another section (Section 4). There, we will also address the question regarding whether the epistemic interest still allows us to denote molecular biology as its own scientific discipline. For now, we will stick with the epistemic interest as the most promising candidate for uniting researchers within a scientific discipline and investigate the epistemic interest of those researchers who, according to a large consensus, have contributed to the emergence of molecular biology as a scientific discipline in the first half and middle of the twentieth century.

### 3.3. The Many Origins of Molecular Biology

Rarely have I encountered a phrase as frequently as “origin(s) of molecular biology” during my research on this topic. Even when it is not directly part of a book’s or paper’s title, the content of hundreds of pieces of writing could easily have this phrase in their title. (It would be easy to provide dozens of references as proof, but for reasons of readability and not blowing up the references section, I refrain from doing so.) This immense interest in the early days and beginning of molecular biology is striking. Not only because it is an interesting topic that deservedly receives a lot of attention. It is particularly striking because each time the story is being told differently—at least from a different perspective or with a different emphasis. Obviously, the reason for this variability is that there is no single origin of molecular biology, there are many. Neither is it just the development of another branch of science, nor the wedging off from another, larger discipline, nor the planned and orchestrated convergence or merging of other separate disciplines. It is a powerful synergy of several tendencies, disciplines, persons, and their development that gradually emerged in the scientific landscape of the twentieth century.

The geneticist H. J. Muller, frustrated with the limitations of X-ray crystallography for viral substances, publicly called upon others: “The geneticist himself is helpless to analyze these properties further. Here the physicist, as well as the chemist, must step in. Who will volunteer to do so?” (cited after [29] (p. 400)). Directly or indirectly, this call reached the right ears. The interdisciplinary efforts and synergetic emergence that ensued can hardly be found in other areas of science; nor can anything comparable easily be found in the history of science. Sometimes, these debates might not have been about the actual history and scientific content but about intellectual territory and the division of financial resources: “The bitterness of many of these discussions reflects the fact that the stakes go far beyond the history of molecular biology, relating to the orientation of current research and its financing” [30] (p. 99). “[T]he discussions surrounding the disciplinary status of molecular biology are as virulent as ever indicating that the stakes are high. What is at issue are intellectual territories and ways of producing knowledge, the distribution of resources and the recruitment of new generations of researchers” [20] (p. 5). This can cast doubt and fuel the debates on the status of molecular biology as a scientific discipline. It also calls into question the classical conceptions of disciplines and their demarcation and development, including other areas of science.

However one wants to frame the relationship between biology and molecular biology, the denotation for both is relatively young. This might be less surprising for molecular biology, but also the name “biology” for the general study and science of life as a demarcated area of science arrived as late as the 19th century (Cf. [31,32,33,34]). Molecular biology fully emerged in the second half of the twentieth century, even though lots of groundwork already happened in the 1930s and 1940s. The origin of the term “molecular biology” warrants special attention. W. Weaver, director for natural sciences in the Rockefeller Foundation (for a historical study on the role of this institution see [35]) proposed the name in 1938 and sketched the field as follows (see [36]; cf. also [37,38,39]): “Among the studies to which the Foundation is giving support is a series in a relatively new field, which may be called molecular biology, in which delicate modern techniques are being used to investigate even more minute details of certain life processes” (cited after [36] (p. 582)).

The usage and scope of this new term were not left uncontested. (This is exemplified in the debate “Molecular biology or ultrastructural biology?” between C. H. Waddington and W. T. Astbury in *Nature* [40,41,42].) The name, nevertheless, stuck—not least because of practical reasons, as F. H. C. Crick described: “I myself was forced to call myself a molecular biologist because when inquiring clergymen asked me what I did, I got tired of explaining that I was a mixture of crystallographer, biophysicist, biochemist, and geneticist, an explanation which in any case they found too hard to grasp” (cited after [43] (p. 390)).

Crick played a very important role in the history of molecular biology (see also Section 3.5.1). For now, however, let us focus on someone who has been frequently credited as the “founding father” or “scientific hero” of molecular biology—and one of its major critics.

### 3.4. Erwin Schrödinger and Erwin Chargaff: Hero and Anti-Hero in the Foundational Myth of Molecular Biology

The title heroes of this section have something in common. Not only their first name “Erwin” and their Austrian background. Both play an important role in the history of molecular biology. The biochemist Erwin Chargaff contributed pioneering work to the biochemical origins of molecular biology. He is now probably best known for his observation on the ratios of bases in DNA, known as “Chargaff’s rules”. In his later days, however, he became an outspoken critic of the routes that science had taken (to the worse, in his opinion). Despite his sometimes-cynical criticism, he is a witty and educated author who is still worth reading today. Reading the texts of this “anti-hero of molecular biology” [44] (p. 97) pays off despite—or even because of this heavy criticism—for biologists and philosophers. Whereas Chargaff became an explicit critic of science, the physicist and Nobel laureate Erwin Schrödinger is held in the highest esteem. In particular, there is a glorified picture about how he contributed to molecular biology. To this day, there are publications and symposia on Schrödinger and his texts (for example [45,46,47]).

The relationship between physics and biology is a special one for several reasons, cf. [33]. Of particular interest for the history of science has been the role that physics played in the emergence of molecular biology (cf. [29] and the literature cited on p. 389). A classical question from the philosophy of science is whether biology could be *reduced* to physics. (This reductionism debate remains a battle ground to this day [48]. Regardless of this debate, the relationship between physics and biology has become a much more fruitful enterprise, looking at how their modes of explanation can be integrated, see for example [49]).

Even more prominently than physics, it is the *physicists* who played an essential role in carrying over their approaches, techniques and views from the field of physics to tackling biological phenomena. Along this line, of significant influence is Schrödinger’s little book *What is life?* [50]. It has been said to “mark the birth of molecular biology” [17] (p. 79). (G. S. Stent calls it “Uncle Tom’s Cabin of the revolution in biology that, when the dust had cleared, left molecular biology as its legacy” [43] (p. 392).) In addition to Schrödinger, also Niels Bohr’s [51] and Max Delbrück’s writings [52] have sometimes been claimed to have had a similar effect. However, Schrödinger clearly left the biggest impact and reputation. His little book *What is life?*, based on public lectures he gave in Dublin in 1943, carries the subtitle “The physical basis of the living cell”. With the constant excuse that he himself is no biologist, he summarizes the textbook knowledge of the time and fuels the expectation that the mechanisms of the living cell will turn out to be based on physical principles. The hope to discover new physical laws is born from some of his formulations (for more nuance, see [53], forthcoming).

With regard to the unmanageable amount of accumulated knowledge, Schrödinger sees the universality of knowledge in danger: “[I]t has become next to impossible for a single mind fully to command more than a small specialized portion of it.

I can see no other escape from this dilemma (lest our true aim be lost for ever) than that some of us should venture to embark on a synthesis of facts and theories, albeit with second-hand and incomplete knowledge of some of them—and at the risk of making fools of ourselves” [50] (p. 1).

What has now been the reason for the great popularity and attractiveness of this book? M. Morange puts it as follows: “Schrödinger presented the new results of genetics in a lively, compelling way—much better than the biologists had. Fifty years later, the book has lost none of its seductiveness: its clarity and simplicity make it a pleasure to read.

A modern molecular biologist would feel quite at home reading Schrödinger’s book” [30] (p. 74). (I can confirm this from personal experience, reading it for the first time as an undergraduate student in one sitting.) F. H. C. Crick said about Schrödinger’s book: “Its main point was one that only a physicist would feel it necessary to make, but the book […] conveyed in an exciting way the idea that, in biology, molecular explanations were just around the corner. This had been said before, but Schrödinger’s book was very timely” (cited after [29] (p. 404)).

F. Jacob’s analysis of the time and spirit make clear what made Schrödinger’s book so “timely”: “After the Second World War, many young physicists were shocked by the military use made of atomic energy. Some of them, moreover, were dissatisfied with the direction taken by experiments in nuclear physics, by their slowness and by the complexity inherent in the use of large machines. They saw in this the end of a science, and looked around for other activities. Some turned to biology with a mixture of anxiety and hope: anxiety, because all they usually knew about living organisms consisted of vague recollections of zoology and botany acquired at school; hope because some of their most celebrated elders pointed to biology as a science full of promise. Niels Bohr saw it as the source of new laws of physics awaiting discovery. Schrödinger also prophesied a new and exhilarating era for biology, particularly in the field of heredity. Just to hear one of the leaders in quantum mechanics asking “What is life?” and then describing heredity in terms of molecular structures, inter-atomic bonds and thermodynamic stability was enough to fire the enthusiasm of certain young physicists and to bestow some sort of legitimacy on biology. Their ambition and interest were limited to a single problem: the physical basis of genetic information” [54] (pp. 259–260).

What truth is there now to all these mythical tales—are they more than mere foundational myths? Indeed, many pioneers of molecular biology have credited Schrödinger’s book as having played a major role in their turn to biology. (“Among those who acknowledged their debt to Schrödinger’s book are M. Delbruck [sic], G. Stent, J. D. Watson, F. Crick, M. F. Wilkins, and S. Benzer” [55] (p. 1071). As was J. D. Watson influenced: “from the moment I read Schrödinger’s ‘What is Life,’ […] I became polarized toward finding out the secret of the gene” [56] (p. 239).) However, the actual role and influence of such eminent authorities have been viewed much less heroic (cf. [44,55,57,58,59,60]; S. Sarkar, despite all criticism, gives a positive outline: “Even if all these criticisms of *What is Life?* were fair, it would still remain to Schrödinger’s credit that he had introduced the idea of the genetic code well before 1953, when the structure of DNA was discovered” [60] (p. 633)). “Perhaps this is a case of the new science’s constructing its own historiography, furnishing itself with such prestigious founders as Niels Bohr and Erwin Schrödinger” [30] (p. 75). “It was characteristic of this early phase of homemade history of science by scientists that it privileged groups and schools that one could easily identify. In particular, it concentrated on the role that physics—theoretical physics—played in the foundation of the new direction in biology. Niels Bohr’s paper ‘Light and life’ as well as Erwin Schrödinger’s book *What is life?* played a central role in this foundational myth” [61] (p. 8).

Even if it was indeed a *foundational myth* (cf. [21,62]), it is hard to brush off the impact of more or less famous physicists on molecular biology: “Physicists played an important role in this change in the form of biological knowledge, by the way they conceived and carried out their experiments. In following Delbrück and asking simple questions of biological objects, they obliged these objects to reply in the same simple language. […] The most important contribution of the physicists was perhaps simply to have been convinced (with a certain dose of naiveté) and to have convinced the biologists that the secret of life was not an eternal mystery, but was within reach” [30] (p. 101).

Despite all that, however, it must not be overlooked that a physical perspective did not and does not exhaust the methodological approaches in biology. This is now also widely accepted by philosophers of science: “It [biology] knows questions and modes of explanation that do not exist anywhere else. A philosophy of science that is exclusively oriented towards physics cannot do justice to biology and thus the entirety of science” [15], my translation.

A potential obstacle in this regard is likely the fact that biology is less suited for idealizations and formalized considerations. Unlike physics, where *ideal gas* assumptions are common, to conceptualize an *ideal organism* does not seem appropriate for all kinds of reasons. The term alone appears to summon the wraiths of vitalism and eugenics. However, there is a deeper problem. Biological regularities are substantially different from physical laws. The latter often claim unrestricted validity for all times and spaces. Biology, on the other hand, deals with a restricted subject area, where generalized claims are regularities and tendencies that are full of exceptions and limitations. By comparing biology to the ideals of physics, one could be tempted to ask with J. J. C. Smart: “Can Biology Be an Exact Science?” [63]. This is also among the reasons why biological theories are rarely expressed in neat mathematical form as those in physics. (Although there have been attempts to formalize and even axiomatize biology [64,65]).

In place of some important equations or axioms that would provide the theoretical foundations of molecular biology, there is a peculiar scheme often referred to as the *central dogma of molecular biology*.

### 3.5. The So-Called “Central Dogma of Molecular Biology”

Another option for establishing molecular biology’s disciplinary identity might be the so-called “central dogma of molecular biology”. After a brief historical view on the unfortunate term “dogma”, I want to investigate whether there is anything “dogmatic” about molecular biology in its current form.

#### 3.5.1. Crick’s Legacy

We owe the scheme behind and term of the “central dogma” to F. H. C. Crick [66,67], in addition to many other more or less problematic insights. When hearing about the “central dogma of molecular biology” for the first time, I thought this was a sort of terminological pun, playing around with terms usually used in theological discussions (cf. [68] (p. 92): “This shorter scheme is called the central dogma, a jokingly ecclesiastical reference by Francis Crick to a summary of bulk cellular information flow.”). Perhaps already containing a grain of truth, some criticism—or something more—might be indicated by this odd choice of words. For that reason, a historical clarification of how the term “central dogma” had been introduced is helpful (see also [8] (pp. 337–338)).

“I called this idea the central dogma, for two reasons, I suspect. I had already used the obvious word hypothesis in the sequence hypothesis, and in addition I wanted to suggest that this new assumption was more central and more powerful. As it turned out, the use of the word dogma caused almost more trouble than it was worth. Many years later Jacques Monod pointed out to me that I did not appear to understand the correct use of the word dogma, which is a belief *that cannot be doubted*. I did apprehend this in a vague sort of way but since I thought that *all* religious beliefs were without foundation, I used the word the way I myself thought about it, not as most of the world does, and simply applied it to a grand hypothesis that, however plausible, had little direct experimental support” [69] (p. 109, original emphasis).

However, something does not simply become a dogma (in its usual meaning) by calling it one. Rather, it takes people who believe in it without question and blindly defend it against any sort of doubt or criticism. Is that something that happened with the so-called *central dogma of molecular biology*? Although often reproduced in textbooks without much context, it is very clear that the underlying assumptions can and have been called into question, criticized, modified and changed multiple times (for example, [70,71,72]; looking at those challenges for the central dogma in detail, however, is beyond the scope of this paper). Overall, these lively debates show the openness of molecular biology, rather than its dogmatism.

#### 3.5.2. What Place do Dogmas Have in Molecular Biology?

Stent [43] distinguishes three phases of molecular biology: a romantic phase (1938–1952, centering around Max Delbrück and the *phage group*), a dogmatic phase (1953–1963, dominated by Watson, Crick and the so-called *central dogma of molecular biology*) and an academic phase (1963 until today?). What characterizes today’s phase of molecular biology? Is it new dogmas? Whereas dogmas have their place in religion, in other areas—and especially science—being dogmatic is a sign of decline and something to be avoided. Does molecular biology have any dogmatic or religious traits? Chargaff calls it “cult-like” [12] (p. 94, my translation), talks of “molecular fundamentalists” [12] (p. 107, my translation) and a “molecular kabbalah” [12] (p. 176). Hausmann talks of “newly converted disciples of molecular biology” [13] (p. 53, my translation). Delbrück calls DNA “unmoved mover” [73] (p. 55).

Fortunately, the life sciences do not appear to be trapped in such a dogmatic predicament. To the contrary, it is the strength of the scientific method that ideas and concepts are up to debate at all times (at least in principle). With sufficient justification, once well-established assumptions will be refuted. There is something known as the academic “mainstream” that one cannot easily object to, and the incentives and publication pressure have made science inherently conservative [74]. However, understood as something that a true believer must not doubt, dogmas have no place in science. (The term “dogma” does repeatedly reappear in contemporary debate when uncritically long-held beliefs are called into question, e.g., [75]).

#### 3.5.3. What Remains of Molecular Biology When the Dogmas Have fallen?

The title of this subsection is a direct reference to P. Bieri’s paper [76], in which he asks: “what remains of analytic philosophy when the dogmas have fallen?” (my translation). As different as these two fields—molecular biology and analytic philosophy—might appear on first (perhaps also second and third) impression, and given that Bieri is comparing two very different academic traditions, his intention in this paper is not too different from mine here. (There are some parallels and complementarity that incentivized me to pursue both studying molecular biology and philosophy in the first place.) In both instances, it is a nuisance when *analytic philosophers* look down on other *continental philosophers* or when *molecular biologists* are disdainful towards colleagues working in *ecology*.

If we take the central dogma and the ensuing research on the relations between RNA, DNA and proteins as indicative for a central part of molecular biology’s research program, we can perhaps best put it with R. Creath like this: “every dogma has its day” [77]. I would even like to adapt his summary to our case of molecular biology here: “For all the difficulties we have noted, [the “central dogma”] is by no means a failure. Instead, it is one of the landmarks of [biologic] achievements of the twentieth century. This is not just because it has been historically influential. Its importance is much deeper than that […].

While [there exist] several non-trivial difficulties with [Crick’s] version of that research program, his strategy as a whole has by no means been refuted. It is doubtful, moreover that it is even possible to refute a strategy […].

In any case, [Crick’s] program is not utterly without antecedents […] [molecular biology] makes sufficient improvement on those antecedents that, as long as we consider [the “central dogma”] as a program rather than a doctrine, we can hope that some way can be found for it to triumph over the difficulties we have noted.

In the meantime we have [several] programs of [biological] research […] As programs of research perhaps neither should be reviled as dogma” (modified from: [77] (pp. 384–385)).

Taken together, we can see that the central dogma is also not suited for ensuring molecular biology’s disciplinary identity in a strict sense. On the contrary, it might be seen as indicative of an area of science without any such strict boundaries. The same observation is underpinned by the diversity of different methods, techniques, figures, and intricate historical interweaving of sciences and scientists we have seen so far. One might thus wonder whether molecular biology might just simply be a mixture of other areas of science.

### 3.6. Is Molecular Biology Just a Mash-Up of Other Different Scientific Disciplines?

Does all this mixing of methods and techniques make molecular biology “a conglomerate, in which biochemistry, structural research, biophysics, genetics, microbiology, immunology, virology and whatever else is mixed together” [2] (p. 34, my translation) or even a “science goulash”? (“[W]hoever violates these boundary lines [between the individual sciences] makes a goulash out of the sciences and becomes, for example, a molecular biologist” [2] (p. 72, my translation)). And what about the rest of (non-molecular) biology? The appearance of “classical” biology has certainly changed through its “molecularization” or “molecular revolution”. However, it might rather be the concept of a demarcated scientific discipline that needs to be called into question than “mourning” other branches of biology that have not disappeared.

From all we have established so far, it is safe to say that regardless of its status as its own scientific discipline, molecular biology has always been a fruitful synergy of different approaches and techniques—and will continue to be so. It is exactly the openness and changes that drive science forward. Which label we put on certain things might be important for organizing departments and curricula, but insisting on strict disciplinary boundaries appears to be obsolete in today’s scientific landscape.

Instead of pitting different fields against each other, perhaps the best path forward is to acknowledge the interdisciplinary character of molecular biology. “In truth the chief feature of molecular biology has been its interdisciplinarity” [78] (p. 511). Along the same lines, it is not surprising and testifies to the interdisciplinary attitude that in a bioinformatics textbook [79] the following is given as a motto: “If nature does not discriminate between physics, chemistry and biology, why should […] the sciences that study it do so?” [80], (p. 3, my translation).

Now that we have established interdisciplinarity as an intrinsic feature of molecular biology—held together by a common epistemic interest that tries to understand biological phenomena by studying the interactions of molecules in and between cells—we can proceed to answer the title question.

### 3.7. Answering the Question: Is “Molecular Biology” a Pleonasm or Denotation for a Discipline of Its Own?

My answer is this: molecular biology has always been more of an interdisciplinary framework and thus not a separate scientific discipline of its own (understood in a *narrow* sense). However, for practical and organizational reasons, for academic research and teaching, and for reasons of a common epistemic interest, it is indeed a subject area that might justifiably be called a discipline in a *wider* sense, acknowledging the vagueness of applying the term loosely.

The truth, once again, lies somewhere in the middle it appears. *Neither* is “molecular biology” a pleonasm that adds nothing to the semantic content of “biology”, *nor* is it a clear-cut and well-delineated denotation for a discipline of its own. With this answer and a distinction between a wider and more narrow conception of what counts as a scientific discipline, I find myself close to R. Olby’s position: (Similarly also: “molecular biology (broadly defined as the set of disciplines looking for molecular explanations and using molecular tools like DNA sequencing and PCR: developmental biology, molecular genetics, genomics, immunology, molecular neurology, etc.)” [81] (p. 2).) “It is this association between structural investigation and genetic mechanisms that gave to the molecular biology of the 1950s its alleged novelty, and justified the claim that here was a new discipline formed out of the fusion of specialism, and quite unlike anything that had preceded it. This *narrow* conception is in contrast to the *broad* conception of the subject held by those responsible for introducing the term ‘molecular biology’ in the 1930s and 1940s” [78] (pp. 503–504, original emphasis).

In sum, it is justified to call molecular biology a scientific discipline due to its underlying epistemic interests to investigate biological phenomena through the molecular interactions within and between cells, the battery of techniques required to do so, and the efforts to advance them, irrespective of any demarcation between the life sciences.

Even in uncontroversial cases that count as scientific disciplines (e.g., mathematics, geology, Egyptology, etc.), they all have a formative history. This applies to other areas and periods of science as well. (This also includes other areas of biology. An interesting comparison could be made to the “modern synthesis” of evolutionary biology and its recent extensions and modifications. Interesting parallels might exist to the inception of this branch of science, according to E. Mayr [82] and more recent scholarship [83]. However, this would be beyond the scope of this paper.) Nevertheless, the tension of the topic at hand is best exemplified in molecular biology: “The [disciplines] that are institutionally […] anchored today are historically grown organizational structures that are not due to the “nature” of things” [25] (p. 237, my translation). Trying to define and defend strict demarcation lines between disciplines is equally unhelpful as putting on blinders and specializing in increasingly smaller and narrower domains. “Excessive specialization can impair the perception of problems that do not fall within the framework of a single [scientific discipline] […]. Working across academic subjects, i.e., intradisciplinary or even interdisciplinary or transdisciplinary research, counteracts this” [25] (p. 238, my translation).

I want to close here with a quotation that nicely summarizes our results so far, as well as pointing out open questions—especially with regards to what makes up the epistemic interest that holds molecular biology together: “Molecular biology, as I hope to show, must be regarded as the result of an extraordinarily complex development that can by no means be described in an adequate fashion through, for example, the fusion of already existing biological disciplines, such as microbiology, genetics, or biochemistry. Nor is it simply another biological discipline supplementing the historically established canon of biological disciplines” [7] (p. 33). “It is therefore not simply a tautological joke when Francis Crick proposed, on “doubtful grounds,” as he admitted with an ironically self-deprecating gesture, that molecular biology “can be defined as anything that interests molecular biologists” [84]” [7] (p. 33). (An interesting suggestion has been made by one of the reviewers, namely, to characterize molecular biology *negatively*, i.e., by referring to anything that does *not* interest molecular biologists. These might include biological phenomena without chemical details, physiological aspects of entire organisms, ecological interactions, etc. I agree that this might often provide clearer disciplinary boundaries in practice).

With a more rigorous analysis of its disciplinary past and trajectory in hand, however, we can very well characterize molecular biology as a scientific discipline via its underlying epistemic interest.

## 4. Molecular Biology: An Interdisciplinary Past and an Open Future

With this result and answer to the title question, we might also ask whether this still applies to molecular biology today and what we can expect for its future. What can we speculate about the future of this area of research called molecular biology? Any predictions and prognosis might be futile, as it is the nature of science not to follow a directional path towards a pre-defined goal. (It is also interesting to revisit Crick’s predictions for “molecular biology in the year 2000” [84].) “What we can guess today will not be realized. Change is bound to occur anyway, but the future will be different from what we believe. This is especially true in science. The search for knowledge is an endless process and one can never tell how it is going to turn out. Unpredictability is in the nature of the scientific enterprise. If what is to be found is really new, then it is by definition unknown in advance. There is no way of telling where a particular line of research will lead” [85] (p. 67, cited after [7] (p. 182)).

One thing appears to me of great importance here. It is not something that only concerns molecular biology but is of particular relevance—that is the large amounts of data that pose their own conceptual problems for the life sciences [86,87]. All the genome, sequence and protein databases and the excessive number of papers being published provide a “rapidly evolving body of knowledge, the accumulation of innumerable incalculable facts that overwhelm the individual’s ability to think and remember” [2] (p. 115, my translation)—thus calling for new methods to deal with them.

In light of these challenges, some turn towards new horizons and claim to see such new approaches—and new disciplines—in the emergence of systems biology and synthetic biology. “New generations”, “scientific revolutions”, “paradigm changes”, the “end” or “death” of molecular biology, or even a “new era” of biology have been announced multiple times (I am also omitting citations here to keep the references to a manageable size). Given the result we have obtained as an answer to the title question, however, we can be cautious in following such claims. When discussing the alleged “death of molecular biology”, M. Morange offers the following claim: “Molecular biology was never a discipline [in a narrow sense] and the apparent signs of death as a discipline—absence of new chairs and journals—is meaningless. It was created as a research area, which is still fully active” [88] (p. 40).

Besides all that, however, I argue that the framework of molecular biology as a discipline (in a wider sense) offers enough space and possibilities for new developments to be included in its epistemic interest. Only if that were not the case, one could hold dogmatism against molecular biology. Even if science (for a number of structural and sociopolitical reasons) has become more conservative [74] and less “disruptive” [89], the accusation of dogmatism would not be justified (see Section 3.5.2). Regardless of single narrow-minded individuals, there are no principal obstacles to trying new approaches in molecular biology. For example, including more of a macroscopic perspective has—despite its reductionist tendencies in the past—never been totally excluded from molecular biology. This is even acknowledged by proponents of such an “evolution of molecular biology into systems biology”: “It is, however, incorrect to state that integrative thinking is new to molecular biology” [90] (p. 1249).

“[S]ystems and synthetic biology, […] are, ultimately, more a continuation and culmination of long-standing research programmes in biology than revolutionary new approaches” [91] (p. S42). Before prematurely claiming scientific “revolutions” or “paradigm shifts”, biology would do well to remember its past. “Although the development of systems biology and synthetic biology cannot be compared to the rise of molecular biology in the 1950s, the changes that these two new fields make to the way that biologists work and to what they consider as proof of value, as well as their possible contribution to reducing the gap between functional biology and evolutionary biology, are important for the discipline as a whole” [92] (p. S53).

Putting on a new disciplinary label would hardly bring any significant innovation or improvement to biology. Instead, I think it would be beneficial to use the term “biology” in a more inclusive, interdisciplinary semantic meaning, as it has already been the case for “molecular biology” from the start. Perhaps the expression “life sciences” is the most inclusive one that embraces the plurality of (sub)disciplines at work. Rigid disciplinary boundaries and habitual adherence to assumptions and principles, of which no one knows exactly when—and for what reason—they became established, have certainly promoted scientific progress less often than those phases where new approaches, methods, and technologies were used to open new scientific horizons. These are lessons one can take from the history and philosophy of molecular biology.

The obscurity and large sizes of scientific projects can be overwhelming and call for particular emphasis on interdisciplinary collaborations. This cannot just be a temporary meeting of representatives of different disciplines in the same room or the publication of articles between the same two book covers without really having reached any understanding and afterwards everyone proceeding in their field as before. To reach a proper understanding and progress, “science needs philosophy” [93].

My aim in this paper was to show that the history and philosophy of science offer important lessons and can contribute to a proper understanding of the disciplinary status of molecular biology as well as its current situation and future developments. I argued that “molecular biology” is neither a pleonasm nor a discipline in a narrow sense but that we should acknowledge its interdisciplinary past and open future. One essential insight in that respect is “that we turn away from the perspective of a more or less well-defined disciplinary matrix of twentieth century biology” [7] (p. 34). That is all the more valid for twenty-first century biology.

In addition, learning from the past about one’s area of research—irrespective of any labels seemingly demarcating separate disciplines—can be informative for contemporary researchers forging their own path, which often might not follow any clear demarcation lines either. Life scientists of the twenty-first century will be well advised to be open to new technologies and experimental techniques from different directions instead of defending any disciplinary territory. In fact, it may be exactly this openness that facilitated molecular biology as a scientific discipline in the first place—properly understood as an epistemic interest to study biological phenomena at the molecular and cellular level—while implementing a diverse set of techniques that allow us to manipulate and study the relevant macromolecules at play. In my opinion, this is what qualified molecular biology as a discipline in the past, still does at present, and will continue to do so in the future.

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
