# Peer review of "“Molecular Biology”—Pleonasm or Denotation for a Discipline of Its Own? Reflections on the Origins of Molecular Biology and Its Situation Today"

_biomolecules, 2023, doi:10.3390/biom13101511_

Round 1
Reviewer 1 Report
The paper is a fair assessment of the historical and philosophical discussions around the question of the status of molecular biology as a discipline. It argues for a broad definition of what counts as a scientific discipline, with a focus on molecular biology’s interdisciplinarity and its openness toward future changes. In addition, it claims that history and philosophy of science should not remain foreign to practicing scientists; on the contrary, that they can contribute to widen their horizon in fruitful ways. The paper can be published as it is, with a few minor corrections stated below.
Specific points:
- All quotations should contain, in addition to the reference, the page number(s)
- Line 290: insert „biology“ after „molecular“
- Burian quotation “Newtonian mechanics/auto mechanics” redundant (lines 244 and 452/53): drop one of them
- Line 455: delete first “and”
Author Response
Thanks to reviewer 1 for their kind feedback! I have implemented all the suggested minor changes:
- the page numbers have been added to the references, where they have been missing (I believe due to some error by the Zotero citation manager; however, to be in line with the journal’s citation style, as also pointed out by reviewer 3, the citation style has now been changed to fit the journal with square brackets and page numbers as an addition in round brackets – anyway, they should be included now).
- missing word in line 290 added.
- redundant lines (451-453) removed.
- redundant word (line 455) removed.
Reviewer 2 Report
The article poses the question of whether molecular biology is a discipline and explores this question using previously published historical studies. The primary framing of the question is theory-centric, i.e. to be a discipline is to have a single overarching theory, and does not engage a number of other viewpoints that are mentioned and cited but not described in depth, e.g. which consider a more institutional or tradition-based view of disciplines. Other relevant views on theory in molecular biology, e.g. Culp and Kitcher 1989 (https://www.journals.uchicago.edu/doi/abs/10.1093/bjps/40.4.459?journalCode=bjps) are not mentioned.
In addition, the importance of being a discipline -- why exactly should one care about this issue today? -- is not clearly stated and defended in the introduction. The introduction should better motivate why the disciplinary status of molecular biology matters. What changes in the world as a result -- do scientists practice research differently, institutions evaluate faculty differently, etc? Right now the article seems to be tacitly assuming being a discipline is a high status or prestigious thing that is desirable, and therefore that we should care, but this is debatable in today's world of interdisciplinary research. The motivation/stakes of the article need clarification up front.
The main body of the article currently reads as a long sequence of quotes from biologists, historians, and philosophers whose authority and relevance are not first clearly established for the reader in the manuscript. The author should synthesize the main information from the quotes and state it in their own words, using quotations only where including the exact language is essential. The current reliance on quotes suggests the author has not mastered the material yet and is leaning on other scholars' words to make their point. Section 2 in particular does not seem essential to the article's main point and can be deleted entirely or summarized as a single paragraph at the start of section 3. This should also help substantially shorten the article's length, which I fear is too long to capture most biologists' interest.
it would strengthen the clarify of the author's argument to present a more systematic summary of existing views about molecular biology as a discipline in section 3.1. Viewpoints such as "ultra-discipline" or "super-discipline" are mentioned but not explained adequately. It's also unclear why we should still be understanding the nature of disciplines in terms of theory given that much of philosophy of science has moved past this narrow viewpoint in the last several decades.
There are a number of type-setting errors, e.g. quotes beginning without capitalization. The author should rephrase comments to avoid double parentheses, for instance "(xyz (abc 2023))". The Nachtigall translation line 204 might be better rendered: "but due to purely practical aspects biology is a closed discipline"
Author Response
Thank you for the critical comments and suggestions!
Theory-centrism is indeed a common limited perspective, which I do criticize in the manuscript – but I now tried to present this more explicitly in the text. Culp and Kitcher is indeed an excellent and most relevant addition to the literature here, which I am gladly adding – despite not wishing to expand the attention on theory too much. As reviewer 2 rightfully observes, this is unfortunately where most philosophical attention has remained with its (rather narrow) scope: the notions of theory, scientific revolutions, and reductionism.
Regarding the question why we should care about the disciplinary status, I have added a couple of remarks in the introduction and conclusion – which should now help introducing and motivating the main question to a better degree. As suggested, putting this upfront should improve the manuscript -thank you!
Section 2 is exactly the motivation part for why history matters (in general), the question about disciplinary identity is now even more and more explicitly motivated in the introduction, p. 2, lines 42 ff., as well as other instances throughout the text (esp. section 5). While the suggestion cut section 2 is an interesting one, I think it is worth stressing the importance of history of science explicitly in its own part of the manuscript. The reasons why it is not another subsection of section 3 is exactly for potential readers from the life sciences who are already convinced that history does (or doesn’t) matter, that they can skip this section and jump into section 3 right away.
I understand that excessive citations might be off-putting for some readers, and I tried to keep a decent balance throughout the text and its revisions – and how they are presented in the manuscript. However, there are still plenty of citations included in the current version, as a conscious choice with the intention of putting a spotlight on either (i) historical documentation presented verbatim or (ii) relevant literature from the history and philosophy of molecular biology that has not yet been available outside the German-speaking world.
I did cut back some citations.
The suggestion to summarize the position “ultra-“ or “super-discipline” and others in a more systematic way in section 3.1 is an excellent one. I have now added a few remarks to do exactly that as briefly as possible (without going into too much detail, as it is not a position I fully endorse) – also in response to other reviewers’ suggestion to make the middle position of a discipline in the wider sense, held together by a common epistemic interest, more explicit and clear - which is now done in many instances throughout the text.
I checked for capitalization issues and fixed them, as well as trying to avoid double parentheses as much as possible, but they could not entirely be avoided in some instances, due to notation requirements. Thank you for the suggested translation improvement of Nachtigall, which I gratefully adapted.
Reviewer 3 Report
General comments:
The author presents an engaging, generally well-written essay that appears to have two major interrelated purposes: (1) to cast light on the disciplinary status (or not) of molecular biology, and (2) to defend the importance of the history and philosophy of science to practitioners of science. Although I much enjoyed the various quotations by and anecdotes about leading molecular biologists, I feel somewhat unsatisfied with how this essay ended. Given the interesting and nuanced buildup, I was expecting some kind of resolution to the issues discussed that was never reached. In my opinion, the author discusses a topic that is interesting to both biologists and philosophers, but does not come up with a clearly articulated viewpoint about what molecular biology actually is. Given that, I have three general recommendations that may help improve the potential impact of this essay for a diverse range of biologists, historians and philosophers (not just molecular biologists):
1) At the end, please provide a synopsis of your conclusions. What are the identifying properties of molecular biology that the author finds most important or useful? Interdisciplinarity? A specific set of tools/methods? A specific philosophical approach, perspective or worldview? Or?
2) A comparative approach may be helpful. How does ‘molecular biology’ compare to other ‘disciplines’ or interdisciplinary domains of science. What areas of science would the author consider as truly ‘disciplinary’ and which as ‘interdisciplinary’? Are there any modern areas of science that are not interdisciplinary? What other areas of science are most similar in conceptual/organizational ‘form’ to molecular biology? Is molecular biology unique in this respect? How does molecular biology compare to other interdisciplinary sciences such as biochemistry, biophysics, ecology, molecular ecology, environmental science, geochemistry, geophysics, etc.
3) Is(are) the origin(s) of molecular biology similar (or not) to that of any other sciences, and does it follow any major kind of scientific advance recognized by historians of science (if not paradigm shifts, as recognized by Kuhn, then what?). Interestingly, Ernst Mayr viewed the origin of neo-Darwinism (i.e., the ‘modern synthesis’ of evolutionary biology) not as a paradigm shift, but as a synthesis of multiple perspectives/approaches/techniques from multiple fields of science, including genetics, taxonomy, ecology and paleontology. Did molecular biology originate in a similar way? Or do we need a new conceptual view of how various scientific domains may originate?
Specific comments:
The author’s affiliation is missing.
L 80: Change “generations” to “generation”.
L290: Insert “biology” after “molecular”.
L 579-584: This is an odd compound sentence that could be broken up into several separate sentences.
References: Please reformat the references (numbers in text, with references listed by number at the end of the paper), as followed by Biomolecules.
Footnotes: I am not sure that the journal permits numbered footnotes (which would conflict with their system of numbering references).
Good overall.
Author Response
Thanks also to reviewer 3!
Making the position clearer is an excellent suggestion (also brought up by other reviewers), and the suggested possible strategies have been a great help in proceeding with making such changes and additions. I decided to follow the path outlined with these options as much as possible, albeit taking some of them to the very end would have probably been beyond the scope of the paper. However, I took helpful inspiration from the outlined recommendations, for which I am very grateful.
A clearer presentation and synopsis of my views is now presented (also due to the improvements suggested by the other reviewers). Listing them all would create a rather confusing list, so I perhaps better refer to the “tracked changes”, which I have implemented throughout the text, where the issues were prevalent and needed to be addressed and developed more explicitly. While I did say (p. 7, line 309, as of writing this reply): “It is difficult to find anything comparable in the history of science.”, I now do mention some thoughts for comparison to the extent I felt confident making any comparison. Many excellent questions raised in the recommendation are leading deeper in debates that I ultimately lead beyond my expertise (e.g. when it comes to the details of the dynamics within the modern synthesis and their most recent developments), but overall the changes should point in the direction indicated and hopefully help in addressing the main concerns raised. Specifically, I hope that the resolution to the issue and a more clearly articulated viewpoint has been reached via the implemented changes – especially with regard to what molecular biology actually is, now more explicitly defined through the common epistemic interest spelled out in the text. That is to say that I think 1) has been improved (I believe); 2) and 3) were very informative and helpful, I acknowledge that there are a number of further issues here that I could only very briefly touch upon in the current text.
Specific comments:
- added missing affiliation information to the template.
- line 80 changed to “generation”.
- line 290 missing “biology” added.
- lines 479-584 split up into several sentences.
Thanks for reminding me of the formatting of the references, I have now updated them to the journal’s style, added missing page numbers; since I did not find any objections against the use of footnotes, I kept them for now – but am happy to reformat them, if required in the editorial process. Thanks for making me aware in any event!
Reviewer 4 Report
I thank Dr. Greslehner for the enjoyable reflections on the origins and status of molecular biology. The thrust of the paper is that under a variety of accounts or criteria for demarcating scientific disciplines, molecular biology (hereafter MB) is best seen as an interdisciplinary effort, but one stable and productive enough to warrant practical treatment as a discipline. Dr. Greslehner further suggests that the discussion motivating this conclusion illustrates the value of historical and philosophical analyses to the practicing scientist.
While I enjoyed the paper, I offer a number of remarks and suggestions below that may help to clarify the author's case, at least to a reader such as myself. These remarks are in no particular order.
* The paper sets up a spectrum with pleonasm on one end and strict discipline on the other. But clearly there is a third dimension or possibility here as evinced by the final characterization of MB in the paper: it may be a label for a loose, malleable, and ephemeral interdisciplinary collaboration like, perhaps, "structural biology".
* Gräfrath's tripartite scheme is provisionally adopted as an account of what constitutes a scientific discipline, and microbiology is found to present an ambiguous case with respect to each element. In particular, it is suggested that although epistemic interest might offer the strongest case, it nonetheless fails to sustain a case for the distinctness of molecular biology. This is ostensibly because the epistemic interest of MB is too diffuse and variable.
I have a number of concerns about this component of the argument, some rhetorical and some substantive. Rhetorically, we are promised at the end of Section 3.2 that Section 5 will survey the epistemic interests of scientists who presently consider themselves molecular biologists. This does not seem to have happened; I find no such discussion in Section 5. Similarly, the remainder of Section 3 is supposed to help us discern the epistemic interests of those "who, according to a large consensus, have contributed to the emergence of molecular [biology] as a scientific discipline in the first half [of the] twentieth century." While the remainder of the section does provide a useful history and recounts the standard hagiography of Schrodinger, much of the discussion is in broad and generic terms, e.g., "minute details of certain life processes." It would be helpful if a summary of apparent epistemic interests were provided at the end of Section 3, even if -- given space constraints -- we had largely to take the author's word for it's accuracy and fairness.
Even were such provided, I would have a substantive worry that the epistemic interests of MB are not as diffuse as Dr. Greslehner suggests. The intersection of the interest of physicists in molecular structure, the informational metaphor (or paradigm, depending one's predilections), and a focus on subcellular processes seems to carve out a domain of interacting macromolecules in cells. That is, the epistemic interest of MB seems centered on a class of material phenomena not well addressed by the techniques and theories of classical chemistry and thermodynamics -- macromolecules typically occur in what the chemist would consider sparse numbers, and interacting in ways that require an understanding of submolecular geometric structure and dynamics. Indeed, most of the battery of techniques mentioned earlier in the paper, like the blots of various cardinal directions, are specialized for assaying entities in this curious range between the molecules of the organic chemist and the solid-state matter of the physicist. Why would "macromolecules and their interactions in and between cells" not capture a core sufficiently well-defined to constitute a discipline?
Or perhaps the epistemic interests of MB as it is practiced are best characterized negatively, i.e., in terms of what does not interest the molecular biologist. For instance, there is little to no interest in whole organisms (most students of MB will never have dissected an organism, raised one in the lab, or observed them systematically in the field) and no interest in the smaller molecules well treated by chemical methods.
In fact, in my experience (my first academic life was in biophysics), disciplines as communities often have negatively characterized boundaries that are far narrower than their theoretical commitments would otherwise suggest. Biophysics would seems to cover a vast seam between the theories and tools of the biologist and those of the physicist -- everything from allometric scaling of metabolism and the thermal properties of feathers to the entropically driven processes of viral injection into a cell. In practice, the field (at least, the community that calls themselves biophysicists and publishes in journal with such in the title) seems to vigorously restrict itself to the molecular. While this is disappointing for a number of reasons, it does seem to give the practice a clearer disciplinary boundary. Molecular biology may be the same way.
* Apropos the previous remark, it is suggestive that there seem to be no programs (undergraduate or graduate) that bill themselves as "molecular biology" programs simpliciter. The term, at least in US education and funding, seems inextricably bound up now with "cell biology". A good example is Alberts, et al _The Molecular Biology of the Cell_, an influential textbook in MB. I would like to know the author's thoughts on this point.
* What distinguishes an interdisciplinary effort from a distinct discipline? Put the other way around, if we allow disciplines to have fuzzy boundaries, what is gained by distinguishing a stable, large-scale, long-running interdisciplinary effort from a discipline?
* As a point of style --- and one which author and editor alike may reject --- I am deeply frustrated by the frequency of one-sentence "paragraphs".
* On p7, I am unclear what point is being made with the quotes on lines 314 to 321 (from Morange and de Chadarevian and Rheinberger).
* I agree with remarks about the potential epistemic hazards of over-specialization and the intellectual pointlessness of drawing sharp boundaries between disciplines. But I don't know that a scientist reading this piece would find the admonitions to embrace interdisciplinarity especially helpful. The structure of scientific research as a social practice seems to preclude interdisciplinary research in any serious way (though it is oft touted by agencies like the US NSF). In reality, the only sort of interdisciplinary work that receives significant funding involves teams of individuals with clear and compelling disciplinary identities. One must classify the effort to fund it, and one must defend the classification -- if I cannot explain why my project is clearly MB, then I cannot get funding from the Division of Molecular and Cellular Biosciences.
* In Section 4.1 on p 12, there are two back-to-back quotes from Crick recounting essentially the same story in slightly different terms. I strongly suggest choosing one and dropping the other; the redundancy breaks the flow and serves no purpose.
* On p 14, the quote from Morange suggests that MB "was also a philosophical view of life". This is an interesting angle to raise, but only if something is said about the content of that view. What were the major metaphysical or epistemic commitments that comprised it? Without telling the reading something about the putative view, the remark does no work in the essay.
* Finally, and perhaps most importantly, I'm having a difficult time seeing how the discussion of the origin and disciplinary status of MB should impact the way in which present and future practitioners of MB ought to understand their various theoretical and methodological commitments. What exactly is it that a practitioner of MB should take away from the fact that MB is best seen as a interdisciplinary effort rather than a strict discipline? In what specific way does the history and philosophy of MB as discussed in this piece offer lessons for the future developments of MB? I realize that Dr. Greslehner urges that this future should abandon a "more or less well-defined disciplinary matrix". But it's hard to see how that's even remotely achievable given the institutional structure of modern science. Is there something substantive one can say about the scientific emphases or direction of the field regardless of disciplinary classifications? Is there some myth or mistake that this investigation brings to the fore? If the case is to be made that such investigations are genuinely relevant to the scientist, there must be some actionable lesson to which to point. If so, I would make it as explicit as possible.
I'm uncertain how to separate the usual orthographic errors or infelicities that every manuscript exhibits (including, no doubt, my own comments on this paper) from what the journal calls "quality of the English language". To be as clear as possible: there are no systematic problems with the English rendering of the author's ideas, only a few typos, omissions, and minor problems with word choice that the author and editor are likely to smooth out in the revision process.
Author Response
Finally, also thank you to reviewer 4!
* The paper sets up a spectrum with pleonasm on one end and strict discipline on the other. But clearly there is a third dimension or possibility here as evinced by the final characterization of MB in the paper: it may be a label for a loose, malleable, and ephemeral interdisciplinary collaboration like, perhaps, "structural biology".
REPLY: Yes, exactly. In the current version of the manuscript, I make this clearer and more explicit – as also prompted by the other reviewers. For instance, section 3.6 now drives this point home more clearly.
* Gräfrath's tripartite scheme is provisionally adopted as an account of what constitutes a scientific discipline, and microbiology is found to present an ambiguous case with respect to each element. In particular, it is suggested that although epistemic interest might offer the strongest case, it nonetheless fails to sustain a case for the distinctness of molecular biology. This is ostensibly because the epistemic interest of MB is too diffuse and variable. I have a number of concerns about this component of the argument, some rhetorical and some substantive. Rhetorically, we are promised at the end of Section 3.2 that Section 5 will survey the epistemic interests of scientists who presently consider themselves molecular biologists. This does not seem to have happened; I find no such discussion in Section 5. Similarly, the remainder of Section 3 is supposed to help us discern the epistemic interests of those "who, according to a large consensus, have contributed to the emergence of molecular [biology] as a scientific discipline in the first half [of the] twentieth century." While the remainder of the section does provide a useful history and recounts the standard hagiography of Schrodinger, much of the discussion is in broad and generic terms, e.g., "minute details of certain life processes." It would be helpful if a summary of apparent epistemic interests were provided at the end of Section 3, even if -- given space constraints -- we had largely to take the author's word for it's accuracy and fairness.
Even were such provided, I would have a substantive worry that the epistemic interests of MB are not as diffuse as Dr. Greslehner suggests. The intersection of the interest of physicists in molecular structure, the informational metaphor (or paradigm, depending one's predilections), and a focus on subcellular processes seems to carve out a domain of interacting macromolecules in cells. That is, the epistemic interest of MB seems centered on a class of material phenomena not well addressed by the techniques and theories of classical chemistry and thermodynamics -- macromolecules typically occur in what the chemist would consider sparse numbers, and interacting in ways that require an understanding of submolecular geometric structure and dynamics. Indeed, most of the battery of techniques mentioned earlier in the paper, like the blots of various cardinal directions, are specialized for assaying entities in this curious range between the molecules of the organic chemist and the solid-state matter of the physicist. Why would "macromolecules and their interactions in and between cells" not capture a core sufficiently well-defined to constitute a discipline?
Or perhaps the epistemic interests of MB as it is practiced are best characterized negatively, i.e., in terms of what does not interest the molecular biologist. For instance, there is little to no interest in whole organisms (most students of MB will never have dissected an organism, raised one in the lab, or observed them systematically in the field) and no interest in the smaller molecules well treated by chemical methods. In fact, in my experience (my first academic life was in biophysics), disciplines as communities often have negatively characterized boundaries that are far narrower than their theoretical commitments would otherwise suggest. Biophysics would seems to cover a vast seam between the theories and tools of the biologist and those of the physicist -- everything from allometric scaling of metabolism and the thermal properties of feathers to the entropically driven processes of viral injection into a cell. In practice, the field (at least, the community that calls themselves biophysicists and publishes in journal with such in the title) seems to vigorously restrict itself to the molecular. While this is disappointing for a number of reasons, it does seem to give the practice a clearer disciplinary boundary. Molecular biology may be the same way.
REPLY: The epistemic interest is in fact the criterium that I think has most merit, which I now state more clearly; I only try to question to what extent that epistemic interest might have changed over time, whether it is still the same interest of those researchers of the early days, or whether there has been a significant change in epistemic interest that today’s molecular biology has either (i) radically changed, (ii) would need or has undergone “scientific revolutions”.
Section 5 then continues to see whether there has been a meaningful change in epistemic interest, insofar as working out the molecular details and mechanisms of the key players of transcription, translation and DNA replication have changed molecular biology, as some would claim, towards a “less reductionistic” perspective, where now the dominant epistemic interest would be to understand biological phenomena through “systems biology” and “synthetic biology”, which allegedly would put a different epistemic interest to the forefront. I call this into question by referring to the broad and encompassing epistemic interest that has been worked out through looking at molecular biology’s history.
I am very grateful that these points were raised by the reviewer, because they highlight that main points were previously inviting the misunderstanding that I would hold the (somewhat vague) epistemic interest against molecular biology as a failure. In fact, I want to stress that this is exactly where the strength of molecular biology and its interdisciplinary history lies, that the epistemic interest to understand biological phenomena on the basis of the molecular components, their interactions, and their effects on higher levels of organization is and has been the defining feature of molecular biology, even though it is neither a very strictly demarcated epistemic interest or scientific discipline. Therefore, by addressing the reviewer’s concerns raised here, I hope the clarity and presentation of my arguments have now been improved in response to the welcome critical comments.
One more specific question raised by reviewer 4: “Why would "macromolecules and their interactions in and between cells" not capture a core sufficiently well-defined to constitute a discipline?” My response would be that this is indeed a very good take and characterization that I would have no strong objections against. The tragedy, again, lies in the details, if we were to try to argue why the same characterization would not hold for biochemistry or cell biology, where exactly to draw the boundaries; if certain techniques could be claimed for one discipline, not the other, etc. I hope that clarifying these points in the text has made the overall argument clearer and more in line with the reviewer’s views. Which I think were already more in line, as it were, but I needed to improve the presentation of my own view. Additions/changes now add discussion in sections 3.2, 3.6 and 5, taking these points into account.
The idea about a negative characterization is also an extremely interesting one. I also agree that this is often how things are decided in practice. However, I was not quite sure where to include this, in addition to the clarificatory changes indicated above. In the end, I decided to mention this idea, crediting the reviewer, in a footnote (22) on p. 11, where I cite Crick’s remark that molecular biology “can be defined as anything that interests molecular biologists”. I figured this would be the best place to introduce this interesting complementary thought.
* Apropos the previous remark, it is suggestive that there seem to be no programs (undergraduate or graduate) that bill themselves as "molecular biology" programs simpliciter. The term, at least in US education and funding, seems inextricably bound up now with "cell biology". A good example is Alberts, et al _The Molecular Biology of the Cell_, an influential textbook in MB. I would like to know the author's thoughts on this point.
REPLY: This is a very interesting point, thank you for raising it! I have in fact undergone at the University of Salzburg, Austria a bachelor's program entitled “molecular biosciences” (Molekulare Biowissenschaften) and a master’s program called “molecular biology” (Molekularbiologie). That the name has disappeared from education programs or merged with other terms, is an intriguing observation. Indeed the textbooks are also an interesting case, in addition to Alberts et al., there is also Lodish et al. with their “Molecular Cell Biology” textbook. I now also include a brief discussion of this in the text (pp. 6-7, fn 6).
* What distinguishes an interdisciplinary effort from a distinct discipline? Put the other way around, if we allow disciplines to have fuzzy boundaries, what is gained by distinguishing a stable, large-scale, long-running interdisciplinary effort from a discipline?
REPLY: I don’t think there is much difference besides the fact that perhaps a discipline is the reification of the effort, something more concrete that one can point their finger at, whereas the effort is less easy to identify. Of course, one my deny that there is such a thing as a “discipline” over and above the effort, but since disciplines are part of everyday speech, that would be a strong take. So, with the reasonable, middle-ground notion of discipline in mind, I think it should be considered as a stable, large-scale, long-running interdisciplinary effort. It is only when trying to defend clear-cut boundaries that things become problematic. In a way, one could say that perhaps the effort could be the precursor to the discipline that will ensue later. But addressing these issues would require more unpacking that I am not sure where to include in the already quite extensive discussion. But I do hope that, at least implicitly, the main issue is somewhat addressed by the changes introduced & explained above, that are intended to better present the reasonable middle-ground position we seem to agree on. Ultimately, it is this take on a discipline and acknowledging its fuzzy boundaries that would make making such a distinction obsolete. Which is why (to also touch upon the last comment here) the main takeaway I would like readers from a scientific audience to have, is to not get hung up by any framing of disciplinary demarcations hindering their efforts, as one of the main lessons one can learn from molecular biology’s disciplinary history and the philosophical debates surrounding it – while at the same time hopefully motivating them to engage in the history and philosophy of their field directly as well.
* As a point of style --- and one which author and editor alike may reject --- I am deeply frustrated by the frequency of one-sentence "paragraphs".
REPLY: I completely agree, in the original formatting, there was some indentation that resulted in a much nicer layout and organization of text. With the template and re-formatting of the text by the journal, this is indeed very unfortunate, and I have fixed those single-sentence paragraphs by merging them with others and thus creating a much more comfortable text flow.
* On p7, I am unclear what point is being made with the quotes on lines 314 to 321 (from Morange and de Chadarevian and Rheinberger).
REPLY: Thank you, I provide now more context for the overall argument and why the stakes in the debate might often be not about the actual content, but the fight over department resources and the like.
* I agree with remarks about the potential epistemic hazards of over-specialization and the intellectual pointlessness of drawing sharp boundaries between disciplines. But I don't know that a scientist reading this piece would find the admonitions to embrace interdisciplinarity especially helpful. The structure of scientific research as a social practice seems to preclude interdisciplinary research in any serious way (though it is oft touted by agencies like the US NSF). In reality, the only sort of interdisciplinary work that receives significant funding involves teams of individuals with clear and compelling disciplinary identities. One must classify the effort to fund it, and one must defend the classification -- if I cannot explain why my project is clearly MB, then I cannot get funding from the Division of Molecular and Cellular Biosciences.
REPLY: This is again a very good observations, and I do acknowledge the practical reasons for organization, funding distribution, etc. Hopefully by stressing more the epistemic interest side of the debate, the reader would take more away from the text than embracing interdisciplinarity - and thus also improving stakes in these practical regards, when the point (or pointlessness) of fights over disciplinary boundaries has been explicitly addressed.
* In Section 4.1 on p 12, there are two back-to-back quotes from Crick recounting essentially the same story in slightly different terms. I strongly suggest choosing one and dropping the other; the redundancy breaks the flow and serves no purpose.
REPLY: Good point. Also to reduce the quantity of citations, I eliminated the second, longer quote and only left the reference in a footnote.
* On p 14, the quote from Morange suggests that MB "was also a philosophical view of life". This is an interesting angle to raise, but only if something is said about the content of that view. What were the major metaphysical or epistemic commitments that comprised it? Without telling the reading something about the putative view, the remark does no work in the essay.
REPLY: Again a very good point. The full quote would continue: “it was also a philosophical view of life, with the ambition to naturalize organisms by the discovery of simple rules and principles.” But I agree that one would either need to further explain that view or abstain from mentioning it. Since the text is already quite long and addressing a large number of issues, I thought that cutting this part of the citation would work best.
* Finally, and perhaps most importantly, I'm having a difficult time seeing how the discussion of the origin and disciplinary status of MB should impact the way in which present and future practitioners of MB ought to understand their various theoretical and methodological commitments. What exactly is it that a practitioner of MB should take away from the fact that MB is best seen as a interdisciplinary effort rather than a strict discipline? In what specific way does the history and philosophy of MB as discussed in this piece offer lessons for the future developments of MB? I realize that Dr. Greslehner urges that this future should abandon a "more or less well-defined disciplinary matrix". But it's hard to see how that's even remotely achievable given the institutional structure of modern science. Is there something substantive one can say about the scientific emphases or direction of the field regardless of disciplinary classifications? Is there some myth or mistake that this investigation brings to the fore? If the case is to be made that such investigations are genuinely relevant to the scientist, there must be some actionable lesson to which to point. If so, I would make it as explicit as possible.
REPLY: This is a crucial question, and one directly the potential impact of the paper on its readership; and a point also raised by reviewer 2. In addition the arguments presented to motivate molecular biologists with the history and philosophy of their area of science (presented in section 2 and elsewhere throughout the text), I put some more motivation for readers into the introduction. The big takeaway for readers should be that fighting over arguably fuzzy boundaries can be pointless arguments, when one is pitted against the other; that one should keep an open mind when working on a scientific problem, to think outside the box, speak with specialists from different areas or working with different techniques. As the scientific progress in molecular biology in the past has often been driven by the introduction or transfer of different experimental technologies, one should invite interdisciplinary collaborations and embrace any chance one can get to learn and implement new technologies when they can help solving a scientific puzzle. Besides the development of new technologies, I believe that we will see much progress by simply applying tools that are already being used in other areas of science for different (yet structurally related) purposes and models. The successful application and transfer of tried and tested models, techniques, approaches from computer science, biophysics, structural chemistry, … you name it.
I briefly include these reasons now in the introduction, they are addressed somewhat abstractly in section 2, but it is absolutely worth spelling things out explicitly, especially towards the end, “when the dust has settled”. I tried to strike a balance between doing so and not adding too much new material, and I hope that the additions – especially the new final paragraph – delivers on this issue.
To repeat from a previous reply:
“Which is why (to also touch upon the last comment here) the main takeaway I would like readers from a scientific audience to have, is to not get hung up by any framing of disciplinary demarcations hindering their efforts, as one of the main lessons one can learn from molecular biology’s disciplinary history and the philosophical debates surrounding it – while at the same time hopefully motivating them to engage in the history and philosophy of their field directly as well.” Ideally, I would like them to see that there are important lessons for their own research and career, while also simply being an interesting academic enterprise in and of itself. Especially when making claims and strides towards future biology, it is crucial to have an overview of the past – including not falling for cheap promises of “scientific revolutions” or “paradigm shifts”, or adhering to any dogmas.
Round 2
Reviewer 2 Report
The manuscript is improved in readability and relevance. I have two main comments:
(1) In re-reading the manuscript this time, I think it makes the most sense as a narrative review of the range of influential views about molecular biology as a discipline, with critical commentary from the author. I suggest this framing better captures how the manuscript prioritizes a wide-ranging discussion of quotes and accounts from other authors rather than emphasizing novel historical research or conceptual arguments.
(2) I think the author still needs to provide clearer signposting at the beginning of each subsection. Most importantly, I don’t understand why the author’s answer to the main question of the paper comes in section 3.6. when the following section 4 presents a whole new topic related to the central dogma and associated viewpoints.
Minor editing of English language required.
Author Response
- Thank you. Indeed, this very much captures the spirit of the paper. Thanks to the critical comments and push back from the reviewers certainly helped in making this more clearly the main narrative of the paper in the revised manuscript. I appreciate the feedback and am glad that the improvements seem to have worked in that respect. Also the changes in response to comment (2) certainly helped in further streamlining this narrative and communicating clearer to the reader what the individual (sub)sections are there for.
- That is a good idea, thanks. I have now restructured the sections such that instead of a section 4 after the main result, the discussion of the central dogma is now part of section 3. This is now the new subsection 3.5, and section 3.7 in which the title question is answered is now the last subsection of section 3. Then the final section 4 only serves a conclusion/outlook/discussion purpose. In addition to the new ordering, I included some smoother transitions to signpost more clearly before or at the beginnings of the (sub)sections what the connection to the previous material is.
Besides some typos spotted by reviewer 3, I have proof-read the entire manuscript myself again to ensure that the remaining “minor editing of English language required” has been taken care of.
Thank you once again for the critical and constructive feedback.
Reviewer 3 Report
In my opinion, this revised manuscript deserves to be published. I only have minor specific suggestions at this stage, as follows:
L 42-45: Restructure sentence as “not only…but also”?
L 51: Change “biolog” to “biology”.
L 51-52: The end of this sentence is dangling and requires completion.
In my opinion, this revised manuscript deserves to be published. I only have minor specific suggestions at this stage, as follows:
L 42-45: Restructure sentence as “not only…but also”?
L 51: Change “biolog” to “biology”.
L 51-52: The end of this sentence is dangling and requires completion.
Author Response
Thank you very much.
The suggested language improvements have been implemented.
Thanks once again for spotting them and the overall feedback on the manuscript.